# Screening of Differentially Expressed Genes Based on the ACRG Molecular Subtypes of Gastric Cancer and the Significance and Mechanism of *AGTR1* Gene Expression

**DOI:** 10.3390/jpm13030560

**Published:** 2023-03-20

**Authors:** Haoran Zhang, Shuman Zhen, Pingan Ding, Bibo Tan, Hongyan Wang, Wenbo Liu, Yuan Tian, Qun Zhao

**Affiliations:** 1Second Department of Thoracic Surgery, The Fourth Affiliated Hospital of Hebei Medical University, Shijiazhuang 050011, China; 2Hebei Key Laboratory of Precision Diagnosis and Comprehensive Treatment of Gastric Cancer, Shijiazhuang 050011, China; 3Second Department of Radiotherapy, The Fourth Affiliated Hospital of Hebei Medical University, Shijiazhuang 050011, China; 4Third Department of General Surgery, The Fourth Affiliated Hospital of Hebei Medical University, Shijiazhuang 050011, China

**Keywords:** gastric cancer, *AGTR1 gene*, Asian Cancer Research Group (ACRG), immune infiltration, invasion and migration

## Abstract

Background: The Asian Cancer Research Group (ACRG) classification is a molecular classification established based on the tissues of gastric cancer (GC) patients in Asia. Patients with different ACRG subtypes differ significantly with regard to treatment response and prognosis, which indicates that the ACRG molecular classification is more valuable than the traditional pathological classification. However, the specific differentially expressed genes (DEGs) and the value of the ACRG molecular subtypes of GC have not been studied in depth. Methods: Through the analysis of the GEO database, the DEGs in GC tissues of different ACRG molecular subtypes were investigated. The expression and mechanism of the screened angiotensin II receptor type 1 (*AGTR1*) gene were bioinformatically analyzed and experimentally verified. The role of *AGTR1* in GC cells was mainly investigated using CCK-8, wound-healing, transwell invasion assays, qRT-PCR, and Western blotting. Results: The bioinformatics results showed the presence of multiple DEGs in GC tissues with different ACRG molecular subtypes. Certain DEGs in GC tissues of different ACRG molecular subtypes have prognostic significance. AGTR1 levels in tumor tissues were significantly higher than in paired paracancerous tissues. The prognosis of GC patients with high expression of *AGTR1* was poor (*p* < 0.05). The *AGTR1* gene in GC samples was associated with the expression of immune pathways and immune checkpoint genes. After modifying *AGTR1* expression in cell lines, cells’ proliferation, invasion, and migration abilities and the expression of related genes changed. Conclusions: There were significant DEGs in GC tissues with different ACGR molecular types, among which the increased expression of *AGTR1* was a molecular feature of MSS/EMT type gastric cancer. Further study found that *AGTR1* was closely related to tumor immune infiltration and invasion and may be a new therapeutic target gene for gastric cancer.

## 1. Introduction

Gastric cancer (GC) is one of the most common malignancies worldwide and the fourth leading cause of cancer death [1,2]. The development of modern medical technology and the continuous improvement in GC treatment strategies have resulted in significant progress in the treatment of GC. However, the cure rate of GC is still very low, mainly because individualized precision treatment is difficult to achieve due to the unclear molecular mechanism. Therefore, accurate staging and classification of GC and identification of its specific molecular markers and therapeutic targets are critical for the diagnosis and treatment of GC.

The Asian Cancer Research Group (ACRG) classification categorizes GC into the following molecular subtypes: microsatellite stable/epithelial-mesenchymal transition (MSS/EMT); microsatellite unstable (MSI); microsatellite stable/tumor protein 53 active (MSS/TP53+); and microsatellite stable/tumor protein inactive (MSS/TP53−). Different molecular subtypes of GC have different survival prognoses. MSI has the best prognosis among these subtypes, followed by MSS/TP53+ and MSS/TP53−. MSS/EMT has the worst prognosis. RNA transcription of the MHL1 gene was observed in the MSI subtype, while CDH1 expression was not observed in the MSS/EMT subtype. In addition, the expression levels of the P21 and MDM2 proteins were further measured to separate TP53+ and TP53− [3]. However, the differential gene expression of patients with different ACRG subtypes is unclear. An in-depth understanding of it is of great value to improve the molecular classification of GC and improve GC diagnosis and treatment.

In our study, we selected GSE62254 (gene expression data for ACRG subtypes of GC) from the Gene Expression Omnibus (GEO) database. We screened for differentially expressed genes (DEGs) in the four subtypes. In addition, we screened 11 core genes that significantly affect the overall survival rate of GC patients if differentially upregulated. The biological functions of these genes were analyzed, and the *AGTR1* gene with significant differential expression and well-defined molecular functions was analyzed.

Most studies on the renin-angiotensin (RAS) system in cancer has focused on RAS inhibitors [4,5,6,7,8,9]. As an important gene in the RAS system, *AGTR1* also plays an important role in various cancers [10,11,12,13,14,15]. One study showed that *AGTR1* promotes lymph node metastasis and invasion in breast cancer by regulating CXCR4/SDF-1a [16]. It has been reported that *LINC00852*/miR-140-3p/*AGTR1* is an important pathway to promote the proliferation and invasion of ovarian cancer cells [17]. In contrast, *AGTR1* is a potential cancer suppressor gene, and the high *AGTR1* expression in lung adenocarcinoma promotes the formation of an antitumor microenvironment. It is associated with a low frequency of MET mutations [18]. However, the relationship between *AGTR1* and GC progression remains unclear. The framework of our research is roughly shown in Figure 1.

## 2. Materials and Methods

### 2.1. Microarray Data

#### Screening of Genes Related to the ACRG Subtypes of GC in the Database

In our study, the gene expression data for ACRG subtypes of GC (GSE62254) were downloaded from the GEO database (http://www.ncbi.nlm.nih.gov/geo/, accessed on 1 July 2021). We mainly used the R limma package to analyze the whole-genome expression profile results and to screen for DEGs. The main steps were as follows. (1) The original data were processed by robust multichip average (RMA) normalization. (2) Differential expression analysis was performed using a *t* test (*p* < 0.05 was considered significant). The inclusion criteria (*p* < 0.05, adjusted *p* value < 0.05, |log2FC| > 2 FC (fold change)) were used to preliminarily screen the DEGs in the four subtypes and for data analysis. (3) The R p heatmap package was used to draw a heatmap of DEG expression.

### 2.2. Gene Ontology (GO) and Kyoto Encyclopedia of Genes and Genomes (KEGG) Enrichment Analysis

In this study, based on the Enrichr database, the clusterProfiler (v3.18.1), an R package, was used to analyze DEGs’ GO and KEGG pathways to determine the most abundant biological pathways and functions associated with DEGs. *p* < 0.05 was the inclusion criterion.

### 2.3. Kaplan-Meier Survival Analysis of Differential Gene Expression in Different GC Subtypes

Survival analysis was performed using the Kaplan-Meier plotter database for prognostic analysis. The relationship between the overall survival of GC patients and the expression levels of key genes was evaluated. *p* < 0.05 was considered significant.

### 2.4. Tissue Specimens and Immunohistochemical (IHC) Staining of Patient Samples

10 pairs of paraffin embedded human MSS/EMT GC and adjacent normal gastric tissues were collected from surgical specimens of the Fourth Hospital of Hebei Medical University. None of the patients received adjuvant chemotherapy. Some tissues were placed in cryopreservation tubes and then quickly frozen in liquid nitrogen for preservation, and some tissues were embedded in paraffin. All patients signed informed consent forms. This study was approved by the Ethics Committee of the Fourth Hospital of Hebei Medical University. Five-micrometer-thick paraffin-embedded tissue sections were used for IHC staining. The slides were dewaxed and incubated with 6% hydrogen peroxide and methanol to inactivate endogenous peroxidase and then with 0.1% pepsin for antigen retrieval. After blocking the sections with a protein-blocking solution, the sections were incubated with an anti-AGTR1 polyclonal antibody (China National Protein Co., Ltd., Beijing, China) and subjected to IHC staining according to the manufacturer’s instructions. All slides used for comparison were processed at the same time and cultured for the same duration. Each section was stained with 3,3-diaminobenzene and hematoxylin. The expression intensity of AGTR1 in cancerous tissues and paracancerous tissues was determined at a magnification of 200×.

### 2.5. The Relationship between AGTR1 and the Genes and Pathways Related to Tumor Immune Infiltration in GC

The data for immune-related genes and their immune pathways were downloaded from the Immunology Database and Analysis Portal (ImmPort, https://www.immport.org/, accessed on 27 September 2021) (release39) database. The protein-protein interaction (PPI) network data for the *AGTR1* gene were downloaded from the Search Tool for the Retrieval of Interacting Genes (STRING, https://string-db.org/, accessed on 2 October 2021) v11.5 database (a search database of known PPIs). The Tumor Immune Estimation Resource (TIMER) database (http://timer.cistrome.org/, accessed on 11 October 2021) was used to analyze the association between genes and immunity. The clusterProfiler was used for the enrichment analysis of DEGs. The GSVA package was used to perform rank normalization on the gene expression value of a given sample. The enrichment score was calculated using the empirical cumulative distribution function to obtain the score of the immune pathway in the sample. Pearson correlation analysis was used to calculate the expression correlation between genes, with 0.05 as the threshold.

### 2.6. The Relationship between AGTR1 and the Immune Cells in Tumor Immune Infiltration in GC

Pearson correlation analysis was used to evaluate the correlation between *AGTR1* expression and immune cells using the TIMER database. The correlation between the *AGTR1* gene and different immune checkpoint genes was calculated in GC samples. In addition, we analyzed the correlation between *AGTR1* expression and cell surface molecules of different types of immune cells. Somatic copy number alterations (SCNA) of *AGTR1* were analyzed.

### 2.7. PPI Network Information

To further explore the molecular mechanism of *AGTR1* in GC, the PPI network information of the *AGTR1* gene was downloaded using the STRING (https://string-db.org/, accessed on 2 October 2021) v11.5 database and constructed. The PPI network was created, the PPI network diagram was drawn, and Cytoscape software was used to enhance the network diagram. The clusterProfiler was used to perform GO and KEGG analysis on the PPI network-related proteins where the *AGTR1* gene is located to predict the functions of *AGTR1* and its related genes and to determine the top 10 GO terms and KEGG pathways.

### 2.8. AGTR1-siRNA Synthesis and Transfection

Transfection of *AGTR1*-siRNA into GC cell lines KATOIII and MKN45. The transfection was performed using Lipofectamine™ 2000 transfection reagent according to the manufacturer’s instructions. *AGTR1*-siRNA and NC-siRNA were designed and synthesized by Shanghai Genechem Biotechnology Co., Ltd. Cell lines in the logarithmic growth phase were cultured in six-well plates. When the cells reached 60–70% confluency, they were transfected for 6 h with Lipofectamine ^TM^ 2000 transfection reagent or siRNA. After the cells were rinsed with phosphate-buffered saline (PBS) solution, they were further cultured in a new complete medium.

### 2.9. Quantitative Real-Time Polymerase Chain Reaction (qRT-PCR) Analysis

TRIzol one-step extraction was used to extract total RNA from pretreated tissues or cells. cDNA was synthesized by reverse transcription using the PrimeScript RT reagent kit (TaKaRa, Dalian, China). Quantitative PCR was performed using the 7500 real-time PCR system (ABI, Waltham, MA, USA). The PCR primers were synthesized by Sangon Biotech (Shanghai, China), and GAPDH was used as the internal control. PCR amplification was performed according to the manufacturer’s instructions, and the relative expression levels of the target genes were quantitatively analyzed using the 2-△△CT method. Primer Sequence used were as follows: AGTR1 F: 5′-GCATTATGTGGACTGAACCG-3′AGTR1 R: 5′-GTGGCTTTGCTTTGTCTTGT-3′ 83bp.

### 2.10. Western Blot Analysis

After 48 h of cell transfection, total proteins were extracted using an extraction kit according to the manufacturer’s instructions. The stacking and separation gels were prepared using the SDS-PAGE gel preparation kit. The samples were loaded on the stacking gel at 80 V (electrophoresis voltage) for 30 min and run on the separation gel at 120 V for 1 h. After SDS-PAGE, the samples were semidry-transferred onto PVDF membranes at 18 V for 20 min. The PVDF membranes were removed and blocked with 5% milk for 2 h and cultured with the following primary antibodies at 4 °C overnight: anti-AGTR1 (FNab00222, Finetest, Wuhan, China), anti-Vimentin (A0326, Abclonal, Woburn, MA, USA), anti-BMP-7 (A0697, Abcam, Cambridge, UK), anti-Smad2 (A11498, Abclonal, Woburn, MA, USA), anti-PD-1 (PRS4067, Merck, Rahway, NJ, USA), anti-VEGFA (SAB5700629, Merck, Rahway, NJ, USA), anti-B-actin (81115-1-RR, Proteintech, Rosemont, IL, USA), anti-E-cadherin (60335-1-Ig, Proteintech, Rosemont, IL, USA), anti-N-cadherin (66219-1-Ig, Proteintech, Rosemont, IL, USA), and anti-GAPDH (ab8245, Abcam, Cambridge, UK). The next day, the membranes were washed 3 times for 15 min in TBST and incubated with secondary antibodies for 2 h at room temperature. Then, the protein bands were developed with an ECL chemiluminescent substrate kit (Biosharp Life Sciences, Hefei, China). Band intensities were quantified using ImageQuant LAS 500 (GE, Boston, MA, USA).

### 2.11. Cell Counting Kit-8(CCK-8) Assay

The cell suspension (100 μL/well) was inoculated into a 96-well plate, and the culture plate was precultured in an incubator (37 °C, 5% CO_2_). A total of 10 μL of CCK solution was added to each well (bubbles in the wells were avoided because bubbles affect the optical density [OD] reading). The culture plate was incubated in the incubator for 1–4 h. The OD at 450 nm was measured using a microplate reader. If the OD value was not measured immediately, 10 μL of 0.1 M HCl solution or 1% *w*/*v* SDS solution was added to each well, and the plate was covered and stored at room temperature in the dark. The OD is stable if measured within 24 h.

### 2.12. Wound-Healing Assay

After the cells in each group reached 100% confluence, a 200-μL pipette tip was used to draw a straight line at the bottom of the well plate. The cells were washed with PBS and cultured for another 24 h. The scratch-healing rate (%) = (1 − (average width of scratches in each group at each time point/scratch width at 0 h)) × 100%.

### 2.13. Transwell Invasion Assay

Transwell chambers (Corning) were placed in 24-well plates. Matrigel gel (Corning Inc., Corning, NY, USA) was diluted to 200 μg/mL, and 150 μL (37 °C) of the diluted Matrigel gel was added to each chamber. After the gel had solidified, MKN45 and KATOIII cells in the logarithmic growth phase were collected and diluted to 5 × 10^5^ cells/mL in serum-free RPMI-1640 and IMDM. The upper chambers were given 150 μL of cell suspension, and the lower chambers were given 700 μL of RPMI-1640 and IMDM complete medium containing 20% fetal bovine serum. After 24 h of incubation in a 37 °C incubator, the chambers were removed and rinsed twice with PBS. The residual Matrigel gel in each chamber was wiped off with a cotton swab. After the chambers were fixed with absolute methanol for 20 min and stained with crystal violet for 5 min, they were placed upside down under a microscope and observed and imaged in the middle, top, bottom, left, and right fields. The experiment was repeated three times.

### 2.14. Statistical Analysis

SPSS 21.0 software was used to analyze the relevant data. The paired measurement data were compared using *t* test, and the count data were compared using the χ^2^ test, with a test level of α = 0.05.

## 3. Results

### 3.1. Screening of DEGs in GC with Different Molecular Subtypes Using Bioinformatics

Through analysis of the GEO database, we found multiple DEGs in GC tissues of different ACRG molecular subtypes (Figure 2A,B). The DEGs in GC tissues of different ACRG molecular subtypes has prognostic significance. A total of 181 meaningful genes were identified through the screening. Among them, 11 genes were selected for differential analysis and prognostic analysis.

### 3.2. GO and KEGG Enrichment Analysis of the Functions of DEGs

ClusterProfiler was used for the enrichment analysis of DEGs and pathways. The analysis results were summarized according to the screening criterion (*p* < 0.05), and the results are shown in Figure 2C. From these enrichment analyses, we observed an association between the DEGs and tumors. It might have provided a breakthrough point for solving the mystery of the formation mechanism of GC.

In the GO biological process (BP) enrichment analysis, the DEGs with the top 10 enrichment scores were significantly enriched in the cellular response to hormone stimulus, regulation of extracellular matrix organization, regulation of cardiac conduction, regulation of heart contraction, and regulation of muscle contraction. “Regulation of extracellular matrix organization” reminded us that the DEGs might be involved in tumor invasion and metastasis. The formation of an extracellular matrix (ECM) is essential for processes such as growth, wound healing, and fibrosis. The ECM’s composition and organization are spatiotemporally regulated to control cell behaviour and differentiation, but dysregulation of ECM dynamics leads to the development of diseases such as cancer. The ECM serves as the scaffold upon which tissues are organized and provides critical biochemical and biomechanical cues that direct cell growth, survival, migration, and differentiation and modulate vascular development and immune function [19]. Dynamic invasion and metastasis of tumors usually involve an extracellular matrix, whose stiffness and elasticity are important for cell migration, gene expression and differentiation.

In the GO cellular component (CC) enrichment analysis, the DEGs were mainly enriched in the sarcolemma, cytoskeleton, collagen-containing extracellular matrix, extracellular membrane-bounded organelle, and extracellular vesicle. The items listed above could tell us that the DEGs were mainly concentrated in the cytoplasm to function. Remarkably, collagen is the main component of the extracellular matrix and determines its stiffness and hardness. Collagens play structural roles and contribute to the mechanical properties, organization, and shape of tissues. In addition, they interact with cells via several receptor families and regulate their proliferation, migration, and differentiation [20,21].

In the GO molecular function (MF) enrichment analysis, the DEGs were mainly enriched in alpha-actinin binding, dipeptidyl-peptidase activity, potassium channel regulator activity, muscle alpha-actinin binding, and sodium channel regulator activity. “Alpha-actinin binding” suggested that these DEGs might be related to cytoskeletal composition, migration, and invasion of cancer cells. Alpha-actinin is a cytoskeletal actin-binding protein and a member of the spectrin superfamily, which bundles actin filaments in multiple cell-type and cytoskeleton frameworks. It forms an anti-parallel rod-shaped dimer with one actin-binding domain at each end of the rod and bundles actin filaments in multiple cell-type and cytoskeleton frameworks [22]. The actin cytoskeleton plays a crucial role in many cellular processes, while its reorganization is important in maintaining cell homeostasis. However, in the case of cancer cells, actin and ABPs (actin-binding proteins) are involved in all stages of carcinogenesis. Furthermore, migration and invasion of cancer cells are based on the formation of actin-rich protrusions (Arp2/3 complex, filamin A, fascin, α-actinin, and cofilin) [23].

In addition, KEGG enrichment analysis was performed to clarify which metabolic pathways and signaling pathways these genes are involved in. The KEGG enrichment analysis results show that the DEGs were mainly enriched in vascular smooth muscle contraction, cGMP-PKG signaling pathway, pancreatic secretion, calcium signaling pathway, and focal adhesion. Calcium signaling pathways are a key second messenger in intra- and inter-cellular signaling pathways. They play an important role in cancer progression, such as sustained cell growth, invasion of other organs, and resistance to cell death inducers [24]. The ability of cells to adhere to the ECM is a fundamental property seen in most multicellular organisms. FA-mediated adhesion of cells to the ECM underlies cell anchorage and cell motility. It allows cells to anchor or migrate, and plays essential roles in development [25], immunity [26], homeostasis [27] and disease [28,29].

### 3.3. Analysis of the Expression Levels of DEGs in Different GC Subtypes

The differential expression analysis of RMA-standardized expression profile data was performed by *t* test. When analyzing one subtype, samples from other subtypes were used as controls. Measurement of the expression values of DEGs (*AGTR1*, *ATG14*, *CDO1*, *CFL2*, *CNN1*, *CNTN1*, *COX7A1*, *DDR2*, *LAYN*, *RASSF8*, and *ZNF471*) in the four subtypes of GC showed that the expression levels of these DEGs were significantly higher in the MSS/EMT subtype than in the other subtypes (Figure 3A).

### 3.4. Kaplan-Meier Survival Analysis of the Expression of DEGs in Different GC Subtypes

The prognostic analysis results using the Kaplan-Meier plotter database indicated that the upregulated expression of 11 DEGs (*AGTR1*, *ATG14*, *CDO1*, *CFL2*, *CNN1*, *CNTN1*, *COX7A1*, *DDR2*, *LAYN*, *RASSF8*, and *ZNF471*) significantly affected the overall survival rate of patients with GC (*p* < 0.05) (Figure 3B).

### 3.5. AGTR1 Expression Verification

We reviewed a lot of literature and found that *AGTR1*, as one of the important coding genes of the RAS system, played an important role in various tumors, but in gastric cancer, there was a lack of in-depth research. The bioinformatics results showed that *AGTR1* was highly expressed in MSS/EMT GC tissues of different ACRG molecular subtypes (Figure 4A). Furthermore, the Kaplan-Meier Plotter database results indicated that the *AGTR1* mRNA level had a significant impact on the OS of GC patients, and the GC patients with high *AGTR1* mRNA expression had a poor prognosis (Figure 4B). Therefore, we speculated that *AGTR1* was an independent predictor of MSS/EMT GC risk factors. Next, we used *AGTR1* as the research object to conduct bioinformatics analysis and experimental verification of its expression and mechanism.

To verify whether *AGTR1* is highly expressed in MSS/EMT GC tissues, we used qRT-PCR technology to recognize the expression of *AGTR1* at the transcriptional level. The results showed that *AGTR1* mRNA expression in GC tissues was significantly higher than in normal tissues (Figure 4C). Furthermore, we measured the expression of AGTR1 in normal gastric epithelial cell lines and GC cell lines using a western blot. As shown in Figure 4C, the expression of AGTR1 in GC cell lines was remarkably higher than that in normal gastric epithelial cell lines. Meanwhile, we performed IHC staining of AGTR1 in 10 pairs of MSS/EMT GC and normal tissues. The results showed that AGTR1 expression in GC specimens was significantly higher than in normal tissues (Figure 4D), which was an effective supplement to our previous bioinformatics analysis results.

### 3.6. The Relationship between AGTR1 and Related Pathways Related to Tumor Immune Infiltration in GC

To understand the relationship more clearly between the *AGTR1* gene and the immune pathways, single sample gene set enrichment analysis (ssGSEA) scoring was performed on the 17 immune pathways of ImmPort in the GC samples, and the correlations of the *AGTR1* gene expression with the immune pathways and immune checkpoint scores were calculated. The results showed that the *AGTR1* gene was associated with most immune pathways in GC samples. Specifically, the *AGTR1* gene was positively related to the TGFb_Family_Member_Receptor, Cytokine_Receptor, and Cytokines pathways but negatively associated with the TNF_Family_Members_Receptors, NaturalKiller_Cell_Cytotoxicity, BCR Signaling Pathway, Interleukins_Receptor, Antimicrobials, and Interleukins (Figure 4E). In addition, the correlations between the *AGTR1* gene and different types of immune checkpoint genes in GC samples were calculated, and it was found that the *AGTR1* gene is associated with immune checkpoint genes such as *TGFB1*, *VEGFA*, *TNF*, *1L4*, *HLA-A*, *PDCD1*, *CD80*, *HMGB1*, and *ENTPD1* (Figure 4F). In summary, this finding suggests that *AGTR1* may play an important role in the occurrence, development, invasion, and metastasis of GC.

### 3.7. Gene Association Analysis of the AGTR1-Related Gene Network and Pathways Using the STRING Database

The PPI network of DEGs was constructed using the STRING database. The more connections that occur between the proteins, the stronger the correlation. 31 proteins, including EDN1/EGFR/ARRB1, were identified and subjected to functional annotation and KEGG enrichment analysis. Under the BP module, DEGs were mainly involved in the cellular response to peptide hormone stimulus, cellular response to peptide, response to peptide hormone, and peptidyl-tyrosine phosphorylation. Under the CC module, the DEGs were mainly concentrated in the membrane raft, membrane microdomain, membrane region, caveola, and plasma membrane raft. In the MF module, the DEGs were mainly enriched in G protein-coupled receptor binding, phosphoprotein binding, cytokine receptor activity, immune receptor activity, and protein-phosphorylated amino acid binding (Figure 4G). KEGG enrichment analysis showed that the DEGs were mainly involved in the cellular response to peptide, cellular response to peptide hormone stimulus, response to peptide hormone, peptidyl-tyrosine phosphorylation, and peptidyl-tyrosine modification (Figure 4H).

### 3.8. Relationship between AGTR1 and a Variety of Immune Cells

To investigate whether *AGTR1* is indeed associated with immune cells, we performed qRT-PCR validation. The results showed that in GC tissues in 10 cases of MSS/EMT GC, *AGTR1* was negatively associated with *PD-1* and positively associated with *VEGFA* (*p* < 0.05) (Figure 5A).

Based on the above validation results, we questioned whether immune cell infiltration is involved in the pathogenic process of *AGTR1* in GC. Therefore, we investigated the relationship between *AGTR1* expression and infiltrating immune cells in GC. Using the TIMER database, it was found that in the immune microenvironment of GC, *AGTR1* mRNA had the most significant correlation to macrophages (cor. = 0.543, *p* = 1.00 × 10^−29^), followed by CD4+ T cells (Cor = 0.379, *p* = 6.06 × 10^−14^) and dendritic cells (cor= 0.321, *p* = 2.35 × 10^−10^). However, in the immune environment of GC tissues, *AGTR1* expression had no exact correlation with CD8+ T cells and neutrophils (Figure 5B).

Analysis of the SCNA of *AGTR1* showed that the various types of immune cells in the samples with different mutation types showed relatively large differences in abundance. Therefore, it was speculated that somatic mutations in *AGTR1* might affect immunity (Figure 5C).

In addition, we also analyzed the correlation between *AGTR1* expression and surface molecules of different types of immune cells. *AGTR1* was significantly positively associated with the expression of several important surface molecules of immune cells. It includes monocyte: CSF1R (cor. = 0.357, *p* = 8.24 × 10^−13^); tumor-associated macrophage (TAM): CCL2 (cor. = 0.421, *p* = 1.09 × 10^−17^) and IL10 protein (cor. = 0.304, *p* = 1.46 × 10^−9^regulatory T-cell (Treg): TGFB1 protein (cor. = 0.449, *p* = 3.53 × 10^−20^) (Figure 5D).

CSF1R, the cellular receptor for Colony Stimulating Factor-1 (CSF1) and Interleukin 34 (IL-34), occupies a central role in manipulating the behavior of TAMs, and the dysregulation of CSF1R signaling has been implicated in cancer progression and immunosuppression in many specific cancers [30]. Small-Molecule CSF1R Inhibitors have been used as Anticancer Agents.

The CCL2-CCR2 axis is one of the major chemokine signaling pathways. It has various functions in tumor progression, such as increasing tumor cell proliferation and invasiveness and creating a tumor microenvironment through increased angiogenesis and recruitment of immunosuppressive cells. CCL2 secreted by cancer cells also has direct cancer-promoting effects [31].

IL-10 is produced by a wide variety of cells, a highly pleiotropic cytokine. Most Tumor-associated macrophages (TAMs) within a tumor release significant amounts of IL-10, which could indicate a correlative rather than a causal relationship between IL-10 and tumor growth. IL-10 might play a dual role in tumorigenesis and development [32].

TGFB1 (transforming growth factor beta 1) is a potent cytokine playing a driving role in development, fibrosis, and cancer.

These results further confirmed the correlation between *AGTR1* expression and the microenvironment of immune cell infiltration in GC.

### 3.9. AGTR1 Expression in GC Cell Lines

To investigate *AGTR1* expression in GC cell lines, we performed qRT-PCR in four GC cell lines (HGC27, NCI-N87, KATOIII, and MKN45) and a gastric epithelial cell line (GES-1). We found that *AGTR1* gene expression in the above four GC cell lines was higher than in the gastric epithelial cell line GES-1 and was the highest in KATOIII and MKN45 cells (Figure 6A). In addition, Western blot analysis showed that AGTR1 protein expression in GC cell lines was significantly higher than in the gastric epithelial cell line (Figure 6B). These results are consistent with our previous bioinformatics analysis and clinical specimen analysis.

### 3.10. Effect of AGTR1 on the Activity, Invasion and Migration of GC Cell Lines

In the cell lines MKN45 and KATOIII, we first verified the protein expression and mRNA after siRNA transfection. The RT-qPCR analysis results are shown in Figure 6C. The mRNA expression level of the transfected group was significantly lower than that of the blank group and that of the NC group (*p* < 0.001). The western blot test in Figure 6D results shows that the protein expression level in the transfected group was significantly lower than in the other two groups (*p* < 0.001), indicating the transfection efficiency of *AGTR1*-siRNA was satisfactory.

To verify the effect of silencing the *AGTR1* gene on GC cell proliferation, we used the CCK-8 assay to measure GC cell proliferation. The results show that after *AGTR1*-siRNA transfection of MKN45 and KATOIII cells, the cell proliferation ability of the *AGTR1*-siRNA group was significantly lower than that of the blank group and that of the negative control group from the second day of culture (*p* < 0.001) (Figure 6E). These results indicated that silencing the *AGTR1* gene could inhibit GC cell proliferation.

We used transwell invasion and wound-healing assays to examine the effect of *AGTR1*-siRNA on the invasiveness of GC cell lines MKN45 and KATOIII. Through the transwell invasion assay, we found that the number of GC cells entering the lower layer in the *AGTR1*-siRNA group was significantly lower than that in the NC group and that the invasiveness of the cells in the *AGTR1*-siRNA group was significantly reduced, which indicated that *AGTR1* could promote GC cell invasion (Figure 6F). In addition, through the wound-healing assay experiment, we found that the migratory ability of the *AGTR1*-siRNA group also significantly decreased (Figure 6G).

In addition, we found through qPCR and Western blot assays that the inhibition of *AGTR1* expression caused a decrease in the mRNA and protein expression levels of VEGFA, N-cadherin, Vimentin, and BMP-7 and an increase in the expression levels of PD-1, E-cadherin, and Smad2 in KATOIII and MKN45 cells (Figure 6H). Therefore, we further believe that *AGTR1* is associated with immunity, invasion, and migration in GC.

## 4. Discussion

The classification of GC has been developing from the initial gross morphological and histopathological classifications to the current molecular classification with the advancement of technology. Classification provides a meaningful guidance basis for the diagnosis and treatment of GC. The Cancer Genome Atlas (TCGA) classification and the ACRG classification are the most influential classification systems of GC [33,34,35]. The ACRG classification considers the correlation between GC subtypes and clinical manifestations and reflects the significant differences in survival time and recurrence rate among different subtypes. Compared with the TCGA classification, the ACRG classification has achieved better results related to GC outcomes. In addition, the TCGA classification was developed based on Asian populations and thus has greater clinical guidance significance for Asian populations [10,36]. GC is a highly heterogeneous disease driven by gene mutations and epigenetic abnormalities. Few drugs are available for GC; targeted drugs are lagging, and GC treatment is still in the bottleneck stage [37]. To date, few studies have been published on screening GC targets based on molecular subtypes. Therefore, we explored the associations between GC targets and molecular subtypes, which may complement and improve the molecular subtyping system.

We screened 181 DEGs in each subtype based on the gene expression data for ACRG subtypes of GC from the GEO database. We performed enrichment analysis and pathway enrichment analysis of these DEGs. The results showed that 11 DEGs had significantly higher expression in the MSS/EMT subtype than in the other subtypes and were associated with a poor prognosis. A literature review showed that *AGTR1* has important effects on tumor growth, angiogenesis, inflammation, and immune function and plays a role in a variety of tumors, such as breast cancer [38,39,40], neuroendocrine tumors [41], and gliomas [42,43,44]. However, the biological functions of *AGTR1* in GC still await exploration. Therefore, we collected clinical specimens to verify the reliability of the bioinformatics results. Next, we constructed a PPI network based on the ImmPort database, the STRING database, and the TIMER database. We found that *AGTR1* was associated with multiple genes and pathways related to tumor immune infiltration in GC. The GO functional annotation and KEGG enrichment of *AGTR1* and related genes mainly manifested as a response to peptide hormone. To determine the association between *AGTR1* and immune cells, we performed experimental verification. *AGTR1* expression was significantly associated with macrophages, CD4^+^ T cells, and dendritic cells. This finding reveals that *AGTR1* may regulate GC through the regulation of macrophages. Macrophages can secrete a variety of cytokines, such as CXCL1, IL6, TGF-β, and VEGF, which can promote tumor growth and metastasis and increase the density of macrophages in the intratumoral region of GC patients. Therefore, it can be inferred that the overexpression of *AGTR1* effectively promotes the immune response and infiltration of macrophages. These findings together indicate that *AGTR1* plays a key role in the regulation and recruitment of immune-infiltrating cells in GC. Finally, we used the CCK-8 assay, wound-healing assay, and transwell invasion assay to confirm that the *AGTR1* gene could promote GC cell migration and invasion. In addition, we were also surprised to find that after the regulation of *AGTR1* expression, the expression of some important targets also changed, which indicates that *AGTR1* is associated with immunity in GC.

It has been reported that the mechanism underlying *AGTR1*-mediated cell motility relies on the activation of the FAK/RhoA pathway. *AGTR1* promotes breast cancer lymph node metastasis by increasing the chemokine CXCR4/SDF-1α and tumor cell migration and invasion [16]. In ovarian cancer, *LINC00852* acts as the ceRNA of miR-140-3p to promote *AGTR1* expression and activate the MEK/ERK/STAT3 pathway, thereby promoting the proliferation and invasion of ovarian cancer cells [17]. In lung cancer, *AGTR1* inhibits the progression of lung adenocarcinoma by promoting the PI3K/AKT3 pathway [10]. In glioblastoma, miRNA-155 regulates *AGTR1* expression after transcription and blocks the *AGTR1*/NF-κB/CXCR4 signaling axis, resulting in a decrease in the levels of *AGTR1* and *CXCR4* and attenuating *AGTR1*-mediated angiogenesis, EMT, stem cell signal transduction, and MAPK signal transduction of tumor cells, which ultimately manifest as reduced tumor proliferation and invasion, reduced lesion formation, and suppressed non-anchored growth [42]. Our study found that *AGTR1* is closely related to MSS/EMT GC. However, the *AGTR1*-mediated pathways have not been thoroughly understood, and the mechanism of action of *AGTR1* in GC development and progression remains to be confirmed by further studies.

## 5. Conclusions

In summary, GC tissues with different ACRG molecular subtypes have significant differential gene expression, and increased *AGTR1* expression is the molecular feature of MSS/EMT GC. Through a more in-depth understanding of the function of *AGTR1*, high *AGTR1* expression can be used as an independent prognostic factor for GC, and *AGTR1* is expected to become an effective tool for the diagnosis and treatment of GC and a new therapeutic molecular target for GC.

## Figures and Tables

**Figure 1 jpm-13-00560-f001:**
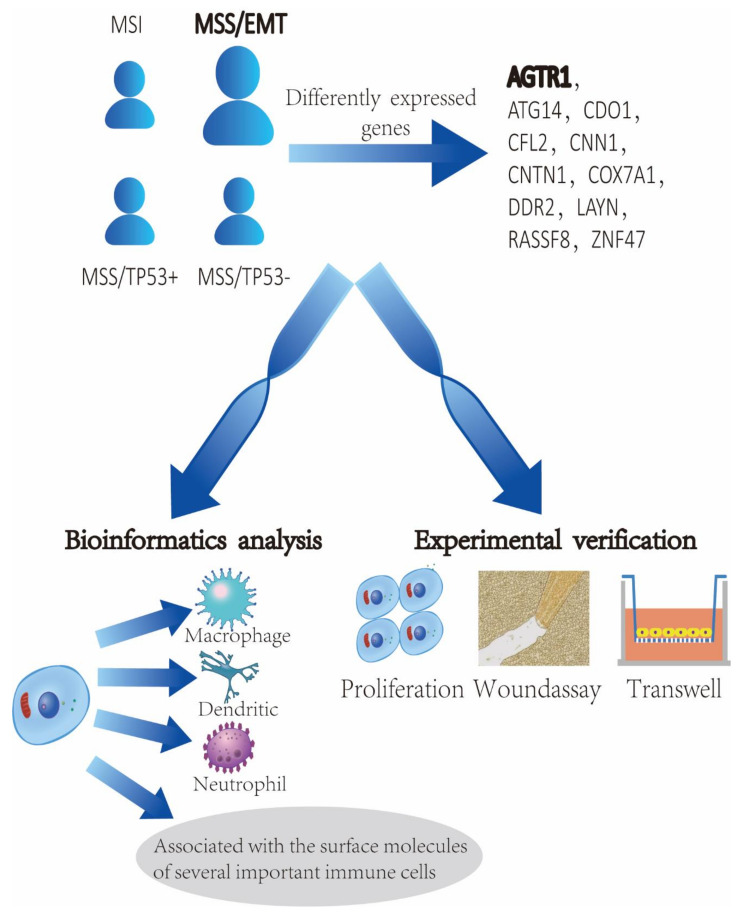
Research Purpose.

**Figure 2 jpm-13-00560-f002:**
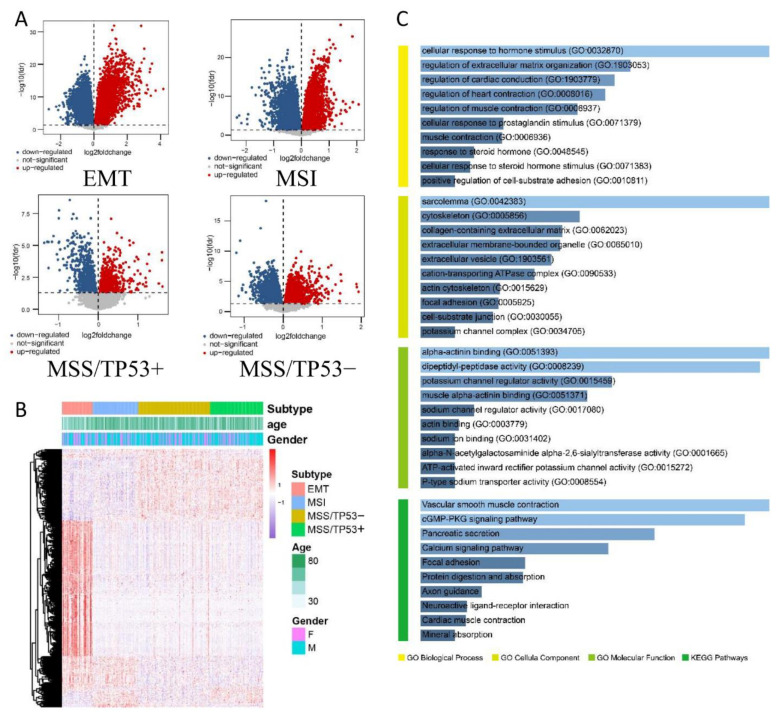
Differential gene expression of different ACRG molecular subtypes in GC. (**A**) Volcano plot of the expression levels of four ACRG molecular subtypes in GC. (**B**) Heat map of DEGs between four molecular subtypes in GC. (**C**) GO and KEGG enrichment analysis. Gene Ontology consists of three modules: biological process, cellular component, molecular function, which reminds us that the DEGs are mainly enriched in extracellular matrix and may be involved in tumor invasion and metastasis. The KEGG enrichment indicated that DEGs are closely associated with calcium signaling pathway and focal adhesion.

**Figure 3 jpm-13-00560-f003:**
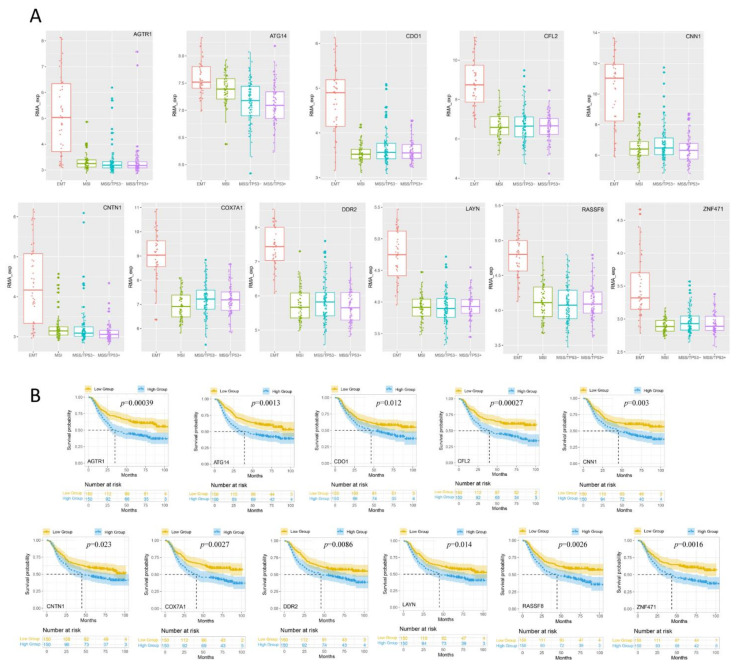
Expression and Prognostic analysis of 11 DEGs in different GC subtypes. (**A**) Box plots of candidate biomarkers (*AGTR1*, *ATG14*, *CDO1*, *CFL2*, *CNN1*, *CNTN1*, *COX7A1*, *DDR2*, *LAYN*, *RASSF8*, and *ZNF471*) expression levels in different GC subtypes. (**B**) Kaplan-Meier survival curves of candidate biomarkers (*AGTR1*, *ATG14*, *CDO1*, *CFL2*, *CNN1*, *CNTN1*, *COX7A1*, *DDR2*, *LAYN*, *RASSF8*, and *ZNF471*) in GC patients based on Kaplan-Meier plotter database.

**Figure 4 jpm-13-00560-f004:**
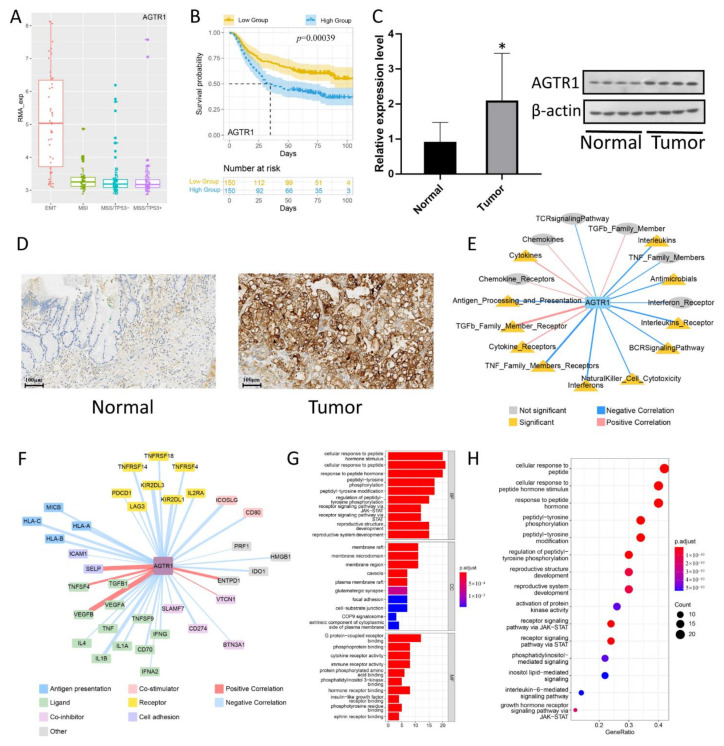
*AGTR1* is related to the expression of immune pathways and checkpoint genes. (**A**) Box plots of *AGTR1* expression levels in different GC subtypes. (**B**) Kaplan-Meier survival curves of *AGTR1* in GC patients. (**C**) Validation of the expression of *AGTR1* in GC Tissues and Cells. (**D**) Representative images of different IHC staining intensities for AGTR1. (**E**,**F**) Relationship between *AGTR1* and related genes and pathways in immune infiltration of GC. (**G**,**H**) GO and KEGG enrichment analysis of the PPI network of *AGTR1*. * *p* < 0.05.

**Figure 5 jpm-13-00560-f005:**
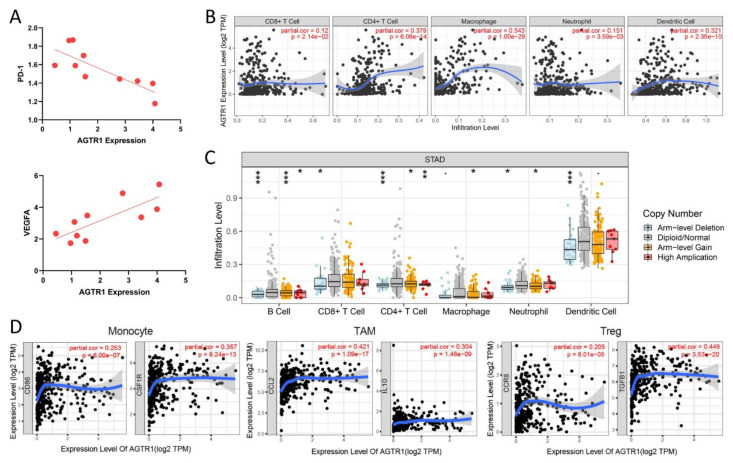
The relationship between *AGTR1* and immune infiltration. (**A**) The correlation between two immune biomarkers (*PD-1*, *VEGFA*) and *AGTR1* was verified through qRT-PCR. (**B**) The relationship between *AGTR1* expression levels and immune cell infiltration in GC via TIMER database. (**C**) Analysis of *AGTR1* somatic copy number alterations. (**D**) Correlation between *AGTR1* expression and surface molecules of different types of immune cells. * *p* < 0.05, ** *p* < 0.01, *** *p* < 0.001.

**Figure 6 jpm-13-00560-f006:**
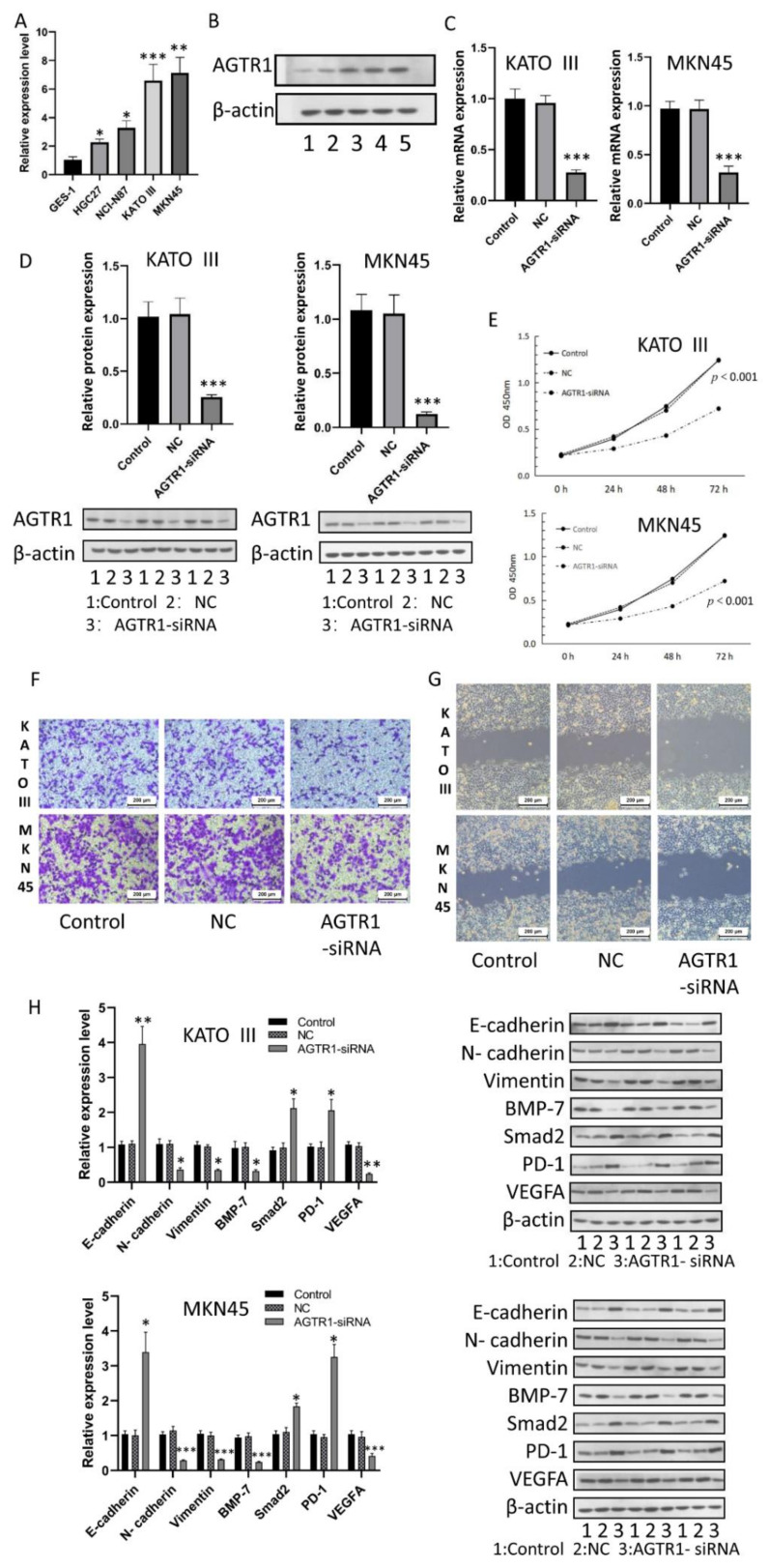
Regulation of *AGTR1* on gastric cancer cell lines. (**A**) The expression level of *AGTR1* was higher than that of GES-1 in four kinds of GC cells (HGC27, NCI-N87, KATOIII, and MKN45). The strongest expression was found in KATO III and MKN45. (**B**) AGTR1 protein expression analysis in GC cell lines. (**C**) The relative levels of *AGTR1*-siRNA mRNA transcripts in KATOIII and MKN-45 cells were analyzed by qRT-PCR. (**D**) Western blot analysis verified the inhibition effect of *AGTR1*-siRNA. (**E**) The proliferation level of GC cells was detected by CCK-8 assay. (**F**) Transwell detected the effect of *AGTR1*-siRNA on the invasion ability of KATOIII and MKN-45 in GC cells. (**G**) The effect of *AGTR1*-siRNA on the KATOIII and MKN-45 migration ability of GC cells was detected by Wound healing assay. (**H**) Effect of *AGTR1* inhibition on the expression of MKN45 and KATOIII marker proteins in gastric cancer cells. * *p* < 0.05, ** *p* < 0.01, *** *p* < 0.001.

## Data Availability

The data that support the findings of this study are openly available in GEO at https://www.ncbi.nlm.nih.gov/geo/query/acc.cgi?acc=GSE62254, accessed on 1 July 2021. We promise that all the data in this article are authentic, valid, and available.

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
