# Peer review of "Screening of Differentially Expressed Genes Based on the ACRG Molecular Subtypes of Gastric Cancer and the Significance and Mechanism of AGTR1 Gene Expression"

_jpm, 2023, doi:10.3390/jpm13030560_

Round 1

Reviewer 1 Report

the subject is very interesting and authors developed it very well. English must be evaluated by a native speakers.

Author Response

Thank you very much for your comments. The paper has been carefully revised and the language has been improved.The style and layout of the figures were also reworked.

Reviewer 2 Report

Zhang et al., using the combination of several bioinformatics tools, suggested a potential clinical value for AGTR1 gene expression in gastric cancer and verified some of their assumptions using tissue samples and in vitro experiments. The number of biological associations was obtained by crossing different data sets with a wide range of backgrounds and data curation, although it appears to be an interesting approach, the noise level and lack of biological connection should be considered within the limitations of this article. . Regarding the experimental assays, there is no information or very few bioinformatics results to support AGTR1 as the relevant gene. The authors did not show sufficient data to support the reduction of DEG from 11 hits to AGTR1, particularly considering the results shown in Figure 3.

On the other hand, the authors claimed validation of their results in silico using tumor tissue and cell lines, however, the authors must provide key technical controls to suggest, at least, verification of their bioinformatics findings.

Important comments:

1) Figure 2A needs labels for each molecular subtype and the results shown in Figure 2C have very limited significance for the study

2) Figure 4 is exactly the same as the inserts in Figures 3A and 3B. The authors performed qRT-PCR for AGTR1 in human tumor samples (Fig 5), so they should add to figure 4C an insert with the mRNA levels for AGTR1 in tumor and non-tumor tissue, confirming that point at the protein and transcription level.

3) In addition, the author must provide the original and unedited version of the digitized immunoblot. The catalog number of each antibody used in this investigation must be included in the respective methods section.

4) Figure 4D needs to be improved, quality is very low, nuclei staining is not shown for tumor tissue. Antibody validation is mandatory, including a control without secondary antibody. The number of cases studied by IHC is not mentioned, as well as the frequency of AGTR1 positivity and the visual cut-off point if available.

5) Staining of immune cells and AGTR1 in cases of tumor FFPE will provide spatial resolution and evidence to affirm the association between AGTR1 and tumor-infiltrating immune cells. That approach will be better than the bioinformatics approach, limiting overfitting.

6) In some figure legends, the authors mentioned “images of the blots cropped from different parts of the same blot were separated by dividing lines”, however, there are no images with that description. Please verify it and add the original blot as supplementary data.

7) Some spelling errors were also detected (Fig. 1, anailisis) (page 16, the text says NCI-N87 but figure 6E says KATOIII)

Reviewer 3 Report

Manuscript ID: jpm-2224819

Screening of differentially expressed genes based on the ACRG molecular subtypes of gastric cancer and the significance and mechanism of AGTR1 gene expression

This article by Haoran Zhang et al. consists of two main parts. First is bioinformatics selection of differentially expressed genes in gastric cancer subtypes and further detailed analysis of one gene selected from them. Next, the Authors performed experimental analyses to verify the reliability of the bioinformatics results. Although the topic is new and of potentially interest, a better presentation of the data and more detailed research design description is necessary that readers can fully profit from this study.

1.       Introduction

-          “Gastric cancer (GC) is one of the most common malignancies worldwide and the third leading cause of cancer death[1, 2].” According to reference no. 2 stomach cancer is fourth leading cause of cancer death.

-          “We screened a total of 11 core genes that significantly affect the overall survival rate of GC patients if differentially upregulated. The biological functions of these genes were analyzed, and the AGTR1 gene with significant differential expression and well-defined molecular functions was analyzed.” The design of the study is based on that hypothesis but the study does not report unequivocal evidence that AGTR1 is key DEGs among the other 10 genes.

2.       Methodology

-              Where is the methodology used in "3.3. Analysis of the expression levels of DEGs in different GC subtypes" described?

-          The description of the "2.9. Quantitative real-time polymerase chain reaction (qRT-PCR) analysis" methods is incomplete, e.g. no primer sequence

3. Results

-         In general, figure labelling, resolution and arrangement need to be improvement. The elements of one figure are divide and presented in different subsections - this needs improvement.

-                   Figure 2: ACRG molecular subtypes are missing.

-          "The analysis results were summarized according to the screening criterion (P < 0.05), and the results are shown in the figure." Figure number is missing.

-                 Figure 3: The authors analysed the ZNF47 or ZNF471 gene?

-          "The results showed that AGTR1 expression in GC specimens was significantly higher than that in normal tissues (Figure 4D), which is consistent with the results of our previous bioinformatics analysis". What previous analyses compared gene expression levels in normal and gastric cancer tissue?

-                 There is almost no description for Figure 4 (E,F and G,H).

-      "Using the TIMER database, it was found that in the immune microenvironment of GC, AGTR1 mRNA had the most significant correlation to macrophages (cor. = 0.543, P = 1.00e-29), significant correlations with CD4+ T cells (Cor. = 0.379, P = 6.06e-14) and dendritic cells (cor. = 0.321, P = 2.35e-10) (P < 0.05), and a correlation with the expression of neutrophils (cor. = 0.151, P = 3.59e-03). In the immune environment of GC tissues, AGTR1 mRNA had no exact correlation with the expression of CD8+ T cells (P > 0.05) (Figure 5B)." The interpretation of the results is unclear. The Authors observed correlation between AGTR1 expression and neutrophils (cor. = 0.151, P = 3.59e-03), but no between AGTR1 and CD8+T cells (cor. = 0.12, P = 2.14e-02). What criteria of significance was adopted?

-          Figure 5: No description for *, **, ***

-          There are some typos in the manuscript, e.g. “anailsis” in Figure 1, GSE-1 or GES-1 cell line etc.

Round 2

Reviewer 2 Report

This reviewer appreciates the added text regarding point 1 and the rationale given by the author for choosing AGTR1.

This reviewer agrees with the author's response to point 2, keep figure 2C and add the additional text in the revised version of this manuscript.

This reviewer appreciates all the changes and adaptations made by the authors in each figure, and in the material and methods section.

Make sure all original immunoblots are available as supplementary data.

Author Response

Please see the revised version of this manuscript.

Reviewer 3 Report

Thank you for Response. I have no more comments.

Author Response

Thank you for your professional review report.